# Cellular Processes Induced by HSV-1 Infections in Vestibular Neuritis

**DOI:** 10.3390/v16010012

**Published:** 2023-12-20

**Authors:** Zhengdong Zhao, Xiaozhou Liu, Yanjun Zong, Xinyu Shi, Yu Sun

**Affiliations:** 1Department of Otorhinolaryngology, Union Hospital, Tongji Medical College, Huazhong University of Science and Technology, Wuhan 430022, China; m202376187@hust.edu.cn (Z.Z.); d202181808@hust.edu.cn (X.L.); u201810264@hust.edu.cn (Y.Z.); sxy2017@hust.edu.cn (X.S.); 2Hubei Province Key Laboratory of Oral and Maxillofacial Development and Regeneration, Wuhan 430022, China

**Keywords:** herpesvirus, herpes simplex virus type 1, vestibular neuritis, celler processes

## Abstract

Herpesvirus is a prevalent pathogen that primarily infects human epithelial cells and has the ability to reside in neurons. In the field of otolaryngology, herpesvirus infection primarily leads to hearing loss and vestibular neuritis and is considered the primary hypothesis regarding the pathogenesis of vestibular neuritis. In this review, we provide a summary of the effects of the herpes virus on cellular processes in both host cells and immune cells, with a focus on HSV-1 as illustrative examples.

## 1. Introduction

Human herpesviruses are large, enveloped, double-stranded DNA viruses that cause a variety of diseases and establish lifelong latent infections in the majority of the global population. The Herpesviridae family comprises nine viruses that are capable of causing human infections and is divided into three subfamilies: Alphaherpesvirinae; Betaherpesvirinae; and Gammaherpesvirinae. Alphaherpesvirinae consists of herpes simplex virus I (HSV-1), HSV-2, and varicella zoster virus (VZV). Betaherpesvirinae includes human cytomegalovirus (HCMV), human herpesvirus 6A (HHV-6A), HHV-6B, and HHV-7. Epstein–Barr virus (EBV) and Kaposi’s sarcoma herpesvirus (KSHV) belong to Gammaherpesvirinae.

Vestibular neuritis (VN) is a clinical condition in the field of otolaryngology characterized by acute and persistent peripheral vertigo caused by unilateral vestibular afferent nerve block. The main symptoms include acute/subacute persistent severe vertigo, accompanied by spontaneous horizontal nystagmus, unsteady posture, and nausea, but without associated auditory dysfunction. The cause of VN is still unknown, but the most common hypothesis is viral infection or reactivation, particularly by HSV (Table 1). Several researchers have reported herpes virus infection in the vestibular ganglion of patients with vestibular neuritis [1,2,3,4,5]. Arbusow et al. reported the presence of HSV-1 DNA in both the human vestibular ganglion and vestibular nuclei [6], suggesting the potential migration of the virus to the human vestibular labyrinth [4]. Furthermore, HSV-1 DNA or HSV latency-related transcripts have been detected in the vestibular ganglia removed during surgery in patients with Meniere’s disease [3,7]. Herpesviruses have a tendency to invade sensory neurons, establish a latency period, and can be reactivated to cause disease. The initial infection or reactivation of the herpesvirus can profoundly affect cellular processes in the host. Mice inoculated with HSV-1 and HSV-2 into the middle ear exhibited hearing loss and vestibular dysfunction. HSV infection was observed in columnar epithelial cells of the infected mice in the stria vascularis, leading to apoptosis in a portion of the infected cells, while many uninfected cells in the spiral organ of Corti also underwent apoptosis. While vestibular ganglion cells did not undergo apoptosis, some of the cells experienced functional loss [8]. Mice inoculated with HSV-1 after auricle also exhibited vestibular dysfunction along with the death of vestibular ganglion cells [9]. In addition to the damage caused to cells by virus infection, the killing of host cells by immune cells may also contribute to the pathogenesis of vestibular neuritis. The coexistence of CD8+ cells and HSV-1 has been observed in the vestibular ganglion cells of patients with vestibular neuritis.

This article provides a summary of the alterations in cellular processes post-infection, using HSV-1 as an exemplars, with the aspiration that this knowledge will aid in the treatment of VN and other diseases instigated by human herpesviruses.

## 2. The Structure of HSV-1 and the Process of Entry into Cells

HSV-1 has a spherical shape [10]. The complete HSV-1 virus comprises double-stranded DNA, a nucleocapsid, teguments, envelope proteins, and a lipid envelope (Figure 1A) [11,12,13]. The nucleocapsid shell exhibits a symmetrical three-dimensional icosahedral structure. HSV-1 is primarily transmitted through close contact. The entry of herpesvirus into human cells for receptor binding and membrane fusion requires the involvement of multifunctional viral glycoproteins on its surface [14,15,16]. HSV-1 carries a minimum of 12 different glycoproteins. There existed four membrane glycoproteins required for HSV entry into cells: the glycoprotein D (gD)-binding receptor; glycoprotein B (gB); and glycoprotein H/glycoprotein L (gH/gL) constitute the core fusion mechanism [17]. Briefly, when HSV is adsorbed on the cell membrane surface, the initial non-specific binding between glycoprotein gC and/or gB and the heparan sulfate mucin (HSPG) on the cell surface reduces the spatial distance between the viral envelope and the cell membrane. gD can specifically bind to herpes virus entry mediator (HVEM), nectin-1, nectin-2, or 3-O-sulfated heparan sulfate (3-OS-HS). This binding further initiates gH and gL. Subsequently, gH-gL transmits signals to gB [18]. gB undergoes a conformational change, inserts into the host cell membrane, and then refolds to fuse the cellular and viral membranes together (Figure 1B). The refolding of multiple gB trimers creates pores in the membrane, initiating the fusion process between the viral envelope and cell membrane. This process may enable the viral nucleocapsid and DNA to enter the cytoplasm and translocate to the nucleus [19].

## 3. The Association between HSV-1 Infection and Vestibular Neuritis

The vestibular ganglion is one of the important sensory nerve structures in the human body, which consists of the superior vestibular nerve and the inferior vestibular nerve. The upper portion of bipolar neurons forms the anterior vestibular nerve, supplying the anterior semicircular canals, otoliths, and part of the utricle. The lower portion of the ganglion cells forms the posterior vestibular nerve, receiving signals from the utricle and posterior semicircular canals. Vestibular neuritis is a type of partial rather than complete paralysis, primarily involving the anterior semicircular canal, horizontal semicircular canal, and utricle [20]. When vestibular neuritis occurs, it can result in complete paralysis of the anterior and horizontal semicircular canals, as well as partial paralysis of the otolith and utricle function. Inflammation that affects the entire vestibular ganglion can lead to complete loss of vestibular function on one side by ocular tilt reaction and spontaneous nystagmus towards the unaffected ear. A case has been reported in which herpes zoster oticus caused facial nerve paralysis, vertigo, and hearing loss on the right side. With the assistance of MRI, the symptoms of the patient are believed to be associated with viral inflammation affecting the right vestibular nerve [21]. Hirata et al. inoculated HSV-1 into the mouse pinna to establish an animal model of vestibular neuritis. Some of the mice infected with HSV-1 showed vestibular dysfunction, and pathological examination revealed degeneration of Scarpa’s ganglion [22]. After inoculation of HSV-1 in the middle ear of mice, all mice showed hearing loss and vestibular dysfunction, suggesting an association between VN and HSV-1 infection [8]. Furuta et al. examined theular ganglia of autopsied adults, and HSV-DNA was detected in 6 out of 10 vestibular ganglia specimens [3]. Arbusow et al. also reported the presence of HSV-1 DNA in human vestibular ganglia and nucleus vestibularis, suggesting that the virus may migrate to the human vestibular labyrinth and cause inflammation [23]. The inflammation in VN patients may lead to acute unilateral vestibular deafferentation and benign paroxysmal positional vertigo. Pollak et al. reported serological evidence indicating a higher prevalence of HSV-1 exposure in patients with VN, which can serve as evidence for HSV-1 infection or reactivation as one of the causes of VN [24]. The vestibular dysfunction observed in the herpes simplex virus labyrinthitis mouse model and the detection of HSV-1 DNA in human vestibular ganglion specimens suggest that vestibular neuritis may be caused by HSV-1 infection or reactivation of latent HSV-1. In fact, it has been demonstrated in vitro that vestibular ganglion neurons can be infected with HSV1 in both a lytic and latent manner [25]. Rujescu et al. reported that genome-wide association analysis revealed an association between the polymorphism rs12979860 and the severity of VN. This rs12979860 locus is associated with the recurrence of herpes simplex and the clearance and treatment outcomes, as well as the severity of the hepatitis C virus [26].

When patients are infected with diseases such as herpes simplex virus (HSV-1) causing herpetic gingivostomatitis, the infected HSV-1 can retrogradely transport along the trigeminal nerve to the vestibular ganglion. The research conducted by Himmelein and colleagues indicates that in 65% of vestibular neuritis, the inferior vestibular nerve was found to travel through two distinct bony canals, unlike the superior vestibular nerve (SVN). Additionally, connections between the facial and vestibulocochlear nerves occurred more frequently with the SVN than with the IVN. This might explain why the superior vestibular is more commonly impacted than the inferior vestibular nerve [2]. When the reactivation of latent HSV-1 in neurons occurs, it leads to selective inflammation of the superior vestibular nerve and typical dysfunction of the vestibular semicircular canals (vestibular paralysis). Moreover, HSV-1 in VN patients may further trigger herpes encephalitis [27].

## 4. Transportation of HSV-1 in Vestibular Ganglion

The transportation of viral capsids and vesicles carrying viral glycoproteins in the cytoplasm is closely linked to microtubules, and their translocation along axons depends on microtubules [28,29,30,31]. HSV-I utilizes microtubules and actin to enter cells retrogradely along axons and undergoes paracrine transport during viral assembly and efflux [28,30,32]. There are two types of axonal transport—fast and slow. Fast axonal transport occurs in both cis and retrograde directions, transporting mitochondria, neurotransmitters, channel proteins, and more [33,34]. In contrast, slow axonal transport occurs in a paracrine direction, transporting cytoskeletal components, such as neurofilaments, microtubule proteins, and actin [35,36]. HSV-1 is actively directed to spread from neurons through the axonal cytoskeleton and molecular motors. Studies using time-lapse microscopy have shown that HSV-1 undergoes rapid axonal flow in both directions [37,38].

## 5. The Replication Process of the Virus within the Cell

After the nucleocapsid is transported to the surrounding area of the nucleus, it can interact with the nuclear pore complex, and then, dsDNA is injected into the nucleus through the nuclear pore [39]. DNA viruses, such as herpesviruses, replicate in specific inclusions within the nucleus. These inclusions, referred to as viral replication compartments (VRCs), are the sites where viral DNA replication, viral transcription, and virion assembly take place [40,41,42]. Compartmentalization is an essential feature of living organisms. Cellular organisms typically utilize cell membranes to partition cells into compartments. Moreover, eukaryotic cells possess membrane-free compartments, such as stress granules and P-bodies [43,44]. Certain compartments exhibit liquid properties and are formed through a process known as liquid–liquid phase separation (LLPS), analogous to the formation of oil droplets in water [45]. There is a hypothesis suggesting that the nuclear VRCs of DNA viruses, such as HSV-1, are also phase-separated condensates [46,47]. Seyffert et al. demonstrated that the HSV-1 transcription factor ICP4 has the ability to induce protein condensation, thereby imparting liquid-like properties to the VRC [48].

After the HSV-1 virus DNA enters the cell nucleus, it is mediated by the host cell RNA polymerase II complex and initiated by the virus capsid protein VP16 to transcribe the viral DNA and initiate the process of viral replication [49,50,51]. VP16, as a capsid protein and a structure of the virus, plays a role in the structural components of the virus once HSV-1 enters the cell. Some cell protein molecules, such as host cell factor 1 (HCF-1) and octamer binding protein-1 (Oct-1), can form multiprotein complexes with viral proteins, binding to the promoter region of the alpha genes in the herpes virus genome and activate the transcription of five alpha genes, namely, ICP0, ICP4, ICP22, ICP27, and ICP47, in a cis-activation manner, thereby initiating the process of linear transcription of the viral genome [52,53,54,55]. Subsequently, beta gene products, such as viral DNA replication, begin to be produced, and the viral genomic DNA molecules start to replicate [55,56,57,58]. When gamma gene products (mostly structural proteins) are sequentially transcribed and translated, the complete replicated gene molecules can be used as the viral genome of progeny viruses for assembly and synthesis [55,59,60].

The process of mature capsids containing viral DNA leaving the nucleus needs the extracellular output complex formed by pUL31 and pUL34. Subsequently, it crosses the nuclear membrane and enters the cytoplasm (Figure 2). During this process, the virus is initially coated with an envelope, which may originate from the inner membrane of the nuclear envelope. Subsequently, the viruses lose the initial envelope through fusion with the outer nuclear membrane and are released into the cytoplasm without an envelope. Upon arrival in the cytoplasm, the capsid is subsequently reenveloped within an intracellular organelle, where it acquires its mature envelope and completes tegumentation. The capsid undergoes secondary envelopment before being released from the cell. During this process, the nucleocapsid, which is now associated with tegument proteins, buds into the membrane of a cytoplasmic organelle, resulting in the formation of an enveloped virion inside a vesicle. The synthesized HSV-1 capsid acquires inner tegument proteins in the cytoplasm, and the outer tegument proteins and viral membrane are obtained through vesicles. The origin of organelle membranes in the secondary envelope is still controversial, with some suggesting that they may originate from membrane tubes derived from recycled endosomes or vesicles from the trans-Golgi network. PUL36 and pUL37 guide the outer shell to move toward the position of the secondary envelope on the microtubule through interaction with the microtubule and also play important roles in the transport of HSV from the nucleus to the periphery by binding to motor proteins. The vesicles contained in gD can effectively transport through axons. Viruses that have completed the secondary envelopment are released through exocytosis.

## 6. The Latency and Reactivation of HSV-1

HSV often results in either lytic or latent infection. After primary infection, HSV-1 can establish a latent state within the neurons of the human sensory ganglia after initial infection. The trigeminal ganglia (TG) serve as the site of latency for HSV-1 in the human body, although the vestibular ganglia, geniculate ganglia, spiral ganglia, and sacral ganglia can also harbor latent viruses [61,62]. During the latency period, the activity of the HSV-1 virus is restricted, only the latency-associated transcripts (LAT) being abundantly expressed. LATs are approximately 8.3 kb long non-coding RNAs expressed in the nucleus of latent infected cells. LAT is involved in establishing latent infection and reactivation processes while also promoting neuronal survival after HSV-1 infection by silencing immediate early gene expression and reducing apoptosis [63,64,65,66,67]. Reactivation can occur spontaneously and be induced by various stimuli. Several viral factors, such as VP16, that may play a role, have been identified; however, the theories and mechanisms underlying latent infection and reactivation are still in need of refinement [68]. The low levels of envelope proteins VP16 and ICP0 may lead to insufficient gene expression, resulting in the latency of HSV-1 in neurons [69]. Infection of neurons by HSV typically results in the production of new infectious viral particles within the cell, whereas infections on axons rarely lead to the synthesis of infectious particles, especially when the viral titer of the infected neurons is low. However, the addition of VP16 to the cell body region of ax infections can result in the production of infectious viruses, indicating that VP16 plays a critical role in the reactivation of the virus [70].

HSV-1 mutants without VP16 transcriptional activation characteristics cannot be effectively reactivated in mice [68,71]. Epigenetic modifications have also been shown to regulate virus latency and reactivation. The DNA of the herpes simplex virus (HSV) lacks histones in its original state. Once it enters the host cell, histone modifications occur on the HSV gene, thereby limiting the expression of important genes [72]. Rujescu et al. identified genome-wide associations with vestibular neuritis in four regions containing protein-coding genes that can be grouped into two functional categories, virus hypothesis and insulin metabolism, through a genome-wide association study [73].

It has been found that certain groups of sensory neurons are more susceptible to latent infection. All neuronal populations in the mouse trigeminal ganglion can be infected by HSV-1; however, neurons positive for KH10 are more susceptible to infection; those expressing the glycosphingolipid-associated embryonic antigen 3 (SSEA3) exhibit a lower rate of infection [74,75]. Similarly, the sensitivity to infection is higher in A5-positive trigeminal and dorsal root ganglion neurons compared to those that are KH10-positive. Flowerdew et al. utilized in situ hybridization to detect latency-associated transcripts in neurons expressing each of the marker proteins [75,76]. The study assessed the frequency of positively labeled neurons and the average size of neurons. Among them, TrkA-positive neurons had the highest frequency, while Ret-positive neurons had the lowest. RT97-positive neurons were the largest in size, while peripherin-positive neurons were the smallest [77].

## 7. Host Cell Processes Caused by HSV-1 Infection

HSV-1 belongs to the lysogenic family of viruses, and its lytic replication results in the destruction of host cells. Cell aggregation is observed almost immediately after cells are infected with HSV-1, and the severity tends to increase with the number of infections [78]. According to Roizman et al. [78], herpes simplex virus infection can lead to the production of multinucleated cells, which result from the fusion of functional cells with different phenotypic characteristics. In HSV-1-infected cells, Avitabile et al. [79] found that microtubules are partially broken, especially at the cell periphery, where the connection between the microtubule network and the plasma membrane appears to be lost. Subsequently, the microtubules form bundles around the nucleus, resulting in a near-spherical shape of the cells [79]. Heeg et al. observed that infection with high doses of various strains of HSV-1 for two and a half hours resulted in cell rounding, accompanied by the breakdown of actin-containing microfilaments and the appearance of knob-like protuberances containing actin at the cell periphery [80]. Hampar et al. reported that HSV-1 infection of cells causes chromosome breaks, translocations, and fusions [81]. Roizman et al. reported that protein synthesis must precede viral DNA synthesis in the early stages of HSV-infected cells [82]. Both functional and structural proteins required for viral proliferation are produced by the host cell’s translation system. HSV has been observed to decrease protein synthesis and mRNA levels in host cells, with the expression level of viral proteins rapidly increasing, accompanied by the rapid degradation of previously existing polyribosomes and some host cell mRNA [83]. Aubert et al. summarize that the manifestations of HSV-1 infection include (i) the loss of matrix binding proteins on the cell surface, leading to detachment; (ii) modifications of membranes; (iii) cytoskeletal destabilizations; (iv) nucleolar alterations; and (v) chromatin margination and aggregation or damage, as well as (vi) a decrease in cellular macromolecular synthesis [78,79,80,81,83,84].

Cellular autophagy, apoptosis, and necrosis pathways are crucial cellular processes that are interconnected to restrict the spread of pathogens by eliminating infected cells [85]. Viral proteins can interact with these signaling molecules, disrupting downstream signal transduction and promoting viral replication and spread. Dufour et al. demonstrated that the ribonucleotide reductase R1 subunit of HSV inhibits Caspase8, thereby protecting cells from apoptosis induced by tumor necrosis factor (TNF) α and Fas ligand [86]. Furthermore, the research group demonstrated that this HSV protein disrupts the structural domain interactions of the Toll interleukin (IL)-1 receptor, thereby inhibiting poly I:C-induced apoptosis in HeLa cells [87]. Moreover, in addition to inducing the formation of filopodia in infected cells to facilitate viral transmission through cell-to-cell contact [88], Us3 proteins disable Bad by inhibiting its phosphorylation [89], thereby safeguarding the cell against DNA fragmentation, nuclear disintegration, and apoptosis [88,90].

Autophagy is a crucial cellular process that involves the self-degradation and recycling of cellular components, including the cell membrane, cytoplasm, and organelles. It plays a role in eliminating misfolded proteins, damaged organelles, and intracellular pathogens. However, certain HSV proteins, such as US11 and ICP34.5, interfere with cellular autophagy. US11 is a ribosome-associated double-stranded RNA-binding protein that directly interacts with PKR [89]. On the other hand, ICP34.5 consists of a C-terminal structural domain and an N-terminal structural domain. The C-terminal domain recruits protein phosphatase 1 (PP-1) to inhibit PKR-mediated phosphorylation of eLF2α [91], while the N-terminal domain directly interacts with Beclin-1 to block autophagy [92]. In summary, there are multiple pathways through which the host induces apoptosis and autophagy in infected cells, and HSV employs its own proteins to interfere with certain steps in these pathways to protect the survival of infected cells and facilitate its own replication and dissemination.

## 8. Immune Cell Process Caused by HSV-1 Infection

The intrinsic and innate immune responses serve as the first line of defense against viral infections, including HSV. They work together to limit the spread of viral replication until the body develops an adaptive immune response. Intrinsic immunity is directly mediated by host cell restriction factors such as promyelocytic leukemia nuclear bodies constituent proteins (PML-NBs) to control viral expression [93]. The innate immune response is initiated through the cellular expression of pattern recognition receptors (PRRs), which detect pathogen-associated molecular patterns (PAMPs) and damage-associated molecular patterns [94]. This recognition stimulates the secretion of interferon (IFN) α, β, or γ, along with other cytokines [95,96,97]. These cytokines can act in an autocrine and paracrine manner and play a crucial role in controlling HSV infection and coordinating innate and adaptive immune responses. Among the PRRs, Toll-like receptors (TLRs) are involved in detecting HSV nucleic acids and proteins (Figure 3). TLR2 interacts with gH/gL on the viral envelope and signals through myeloid differentiation factor 88 (MyD88) [98,99]. TLR2 activation promotes the expression of pro-inflammatory cytokines, exerting antiviral effects. However, studies on TLR2-deficient mice infected with HSV have shown that these mice exhibit fewer symptoms and longer survival than wild-type mice, suggesting that TLR2 activation may have harmful effects on the host [100,101]. TLR3 recognizes dsDNA and induces nuclear factor-kappaB (NF-κB) activation and IFN production to exert antiviral effects [102]. Herpes simplex virus encephalitis can be caused by defects in the TLR 3 pathway [103]. Mouse experiments suggest that astrocytes rely on TLR3 to mediate resistance to HSV infection [104]. However, another study demonstrated that TLR3-deficient neurons and oligodendrocytes were more susceptible to HSV-1 infection compared to control cells, indicating the importance of TLR3 in protecting neuronal cells from HSV infection [105]. TLR9 recognizes HSV DNA and is significant for certain cell types, such as plasmacytoid dendritic cells (pDC), where the absence of TLR9 results in impaired IFN responses [95,106]. The HSV DNA within the cytoplasm is encapsulated by capsid proteins, thereby being protected. However, a mechanism observed in macrophages involves the degradation of the HSV capsid, resulting in the release of HSV DNA into the cytoplasm. Consequently, there are pathways in the cell for the detection of HSV-1 DNA. Cyclic GMP-AMP synthase (cGAS) and IFI16 can detect the released HSV DNA in the cell and activate STING, leading to the recruitment of TBK1, activation of IRF3, and induction of IFN [107,108,109,110].

The adaptive immune response plays a crucial role in managing HSV infection and reactivation. Cell-mediated immunity, particularly involving T cells, is a key component of the adaptive immune response. After viral infection, cells present antigens to CD8+ T cells through surface major histocompatibility complex (MHC) class 1 molecules. This triggers the elimination of infected cells, limiting viral spread. T cells have been found to play a major role in the adaptive immune response to HSV. Specific T cells have been identified in the sensory ganglia of infected individuals and in active and latent lesions of patients [111,112,113,114]. Following acute HSV infection, the percentage of blood-specific T cells is lower in infected individuals [115,116]. HSV-specific CD8+ T cells in the blood express high levels of cytolytic molecules when re-exposed to viral antigens [117]. CD4+ T cells recognize HSV-1 proteins and express cytokines associated with helper T cell type 1 (Th1)/Th0-like responses with cytolytic potential [116,118].

HSV-1 is capable of establishing a latency period in the dorsal root ganglia (DRG) of severely combined immunodeficient mice, even when CD8+ memory T cells are transplanted prior to infection. However, the presence of T cells reduces the number of infected DRG neurons, potentially limiting HSV-1 reactivation [119,120]. In mouse models, the rate of in vitro reactivation of trigeminal ganglia (TG) is directly correlated with viral ganglionic load rather than the number of specific CD8+ T cells [121]. Specific CD8+ and CD4+ T cells are also present in the TG following human HSV-1 infection [111,112]. The infiltrating T cells in human-infected TGs are characterized as memory effector T cells and surround the cell bodies and axons of neurons [111,122]. In mouse models, memory CD8+ T cells express interferon-gamma (IFN-γ), which prevents HSV replication in neurons and inhibits neuronal apoptosis, potentially promoting the survival of neurons and HSV-1 silencing and latency [123,124,125]. The mechanism of CD4+ and CD8+ T cell recognition of latently infected neurons is not fully understood. It is possible that there may be limited viral gene expression that can be recognized by T cells, allowing for the CD8+ T cell recognition and reactivation, along with potentially low levels of neuronal MHC class I molecule expression [126]. Additionally, satellite cells can act as antigen-presenting cells and express T-cell suppressor molecules to control HSV-1 latency without damaging neurons [127]. HSV also employs various strategies to inhibit antigen presentation and modulate adaptive immune responses. For example, the viral protein ICP47 blocks antigen presentation, and ICP34.5 inhibits autophagy, which is involved in antigen presentation [128]. Furthermore, HSV can inhibit antibody responses by interacting with antibodies and complement components, inhibiting antibody-dependent cell-mediated cytotoxicity [129]. These mechanisms suggest that HSV can modulate the adaptive immune response and influence the pathogenesis of the infection.

## 9. Conclusions

The presence of herpesviruses has been detected in surgically removed tissues from patients with Meniere’s disease, and viral infection is considered a leading hypothesis for the development of vestibular neuritis. However, the exact role of herpesviruses in the pathogenesis of vestibular neuritis is still not fully understood. In this paper, we aimed to summarize the effects of herpesvirus infections, particularly HSV-1, on host cells and immune system processes. This information can be valuable in predicting neuronal cell damage, as well as the infiltration and killing of immune cells following viral infection. We hope that this review will stimulate further work and efforts to advance the prevention and treatment of diseases like vestibular neuritis that are potentially caused by viral infections.

## Figures and Tables

**Figure 1 viruses-16-00012-f001:**
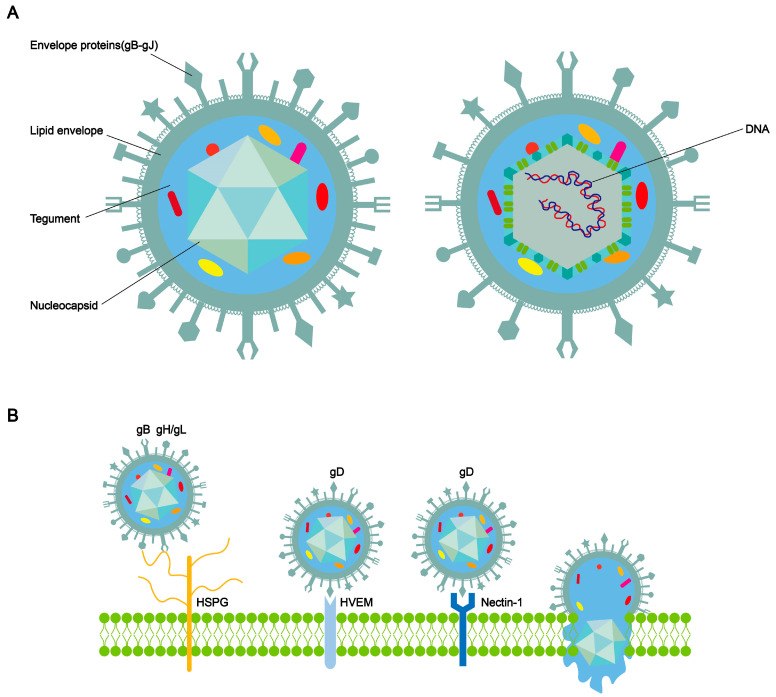
(**A**) Schematic diagram of the structure of HSV-1. HSV-1 is composed of double-stranded DNA, nucleocapsid, tegument, envelope proteins, and a lipid envelope. (**B**) Schematic diagram of HSV-1 entry into cells via envelope proteins. While the types and composition of envelope proteins may vary among different viruses in the sporozoan virus family, the fusion of the viral envelope with the host cell membrane and the entry of the nucleocapsid rely on the binding of multiple envelope proteins to receptors on the host cell membrane and conformational changes.

**Figure 2 viruses-16-00012-f002:**
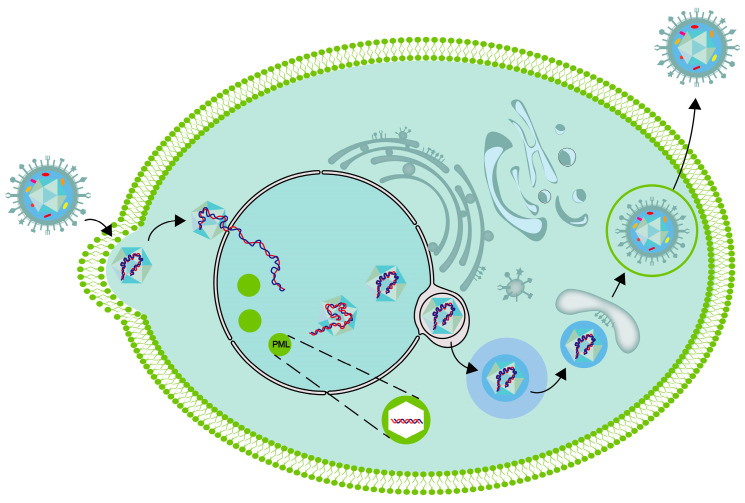
Schematic diagram of intracellular cycle of HSV-1.

**Figure 3 viruses-16-00012-f003:**
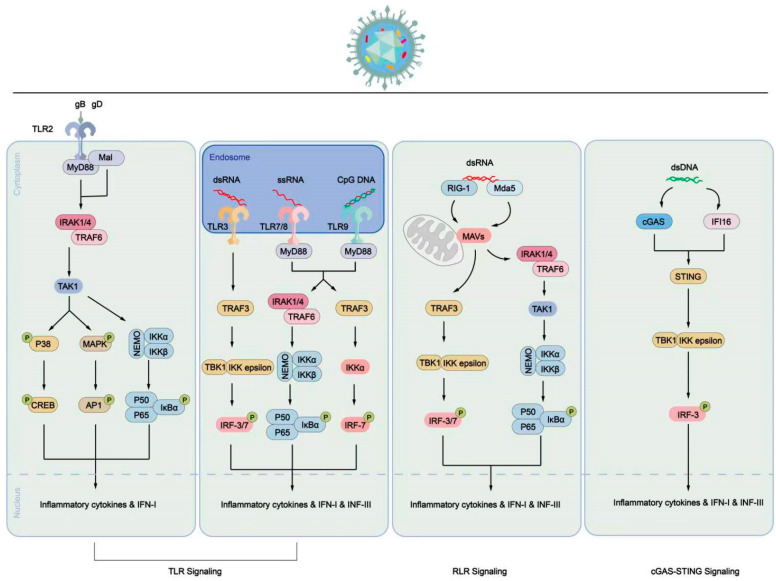
Pattern diagram of immune cell response process triggered by PRR signal triggered by HSV-1 infection. Inducing the secretion of inflammatory cytokines or IFN through the TLR signaling pathway (left, middle) or RLR signaling pathway (right). In the TLR signaling pathway, TLR2 recognizes signals induced by HSV-1 envelope proteins, such as gB or gD. The signal is transmitted to the cytoplasm, where MyD88 binds to the cytoplasmic domain of TLR2, leading to the activation of transcription factors like NF-κB. This activation promotes the translocation of P50/P65 into the nucleus and increases the expression of inflammatory cytokines and IFN-1. Additionally, TLR3, TLR7/8, and TLR9 signaling are activated by dsRNA, ssRNA, or CpG DNA, respectively, in endosomes. These signals activate IRF-3, IRF-7, and NF-κB, ultimately resulting in increased expression of inflammatory cytokines, IFN-1, IFN-III, and interferon-stimulated genes (ISGs). In the RLR signaling pathway, RIG-I and MDA5, which contain N-terminal caspase activation and recruitment domains, recruit and activate the mitochondrial antiviral signaling (MAVS) protein to mediate signal transduction. The activated MAVS protein further activates downstream signaling, promoting the expression of inflammatory cytokines and IFN. Both pathways contribute to the immune response against HSV-1 infection by triggering the production of inflammatory cytokines and interferons, which play crucial roles in controlling viral replication and coordinating innate and adaptive immune responses.

**Table 1 viruses-16-00012-t001:** Viruses associated with VN.

Viruses Associated with VN	Family of Viruses
Herpes simplex virus	Alphaherpesvirinae
Varicella-zoster virus	
Human cytomegalovirus	Betaherpesvirinae
Epstein–Barr virus	Gammaherpesvirinae
Influenza virus A	Non-herpesvirus family
Influenza virus B	
Adenoviruses	
Rubella virus	
Parainfluenza virus	

## Data Availability

Not applicable.

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
