# Peer review of "Cellular Processes Induced by HSV-1 Infections in Vestibular Neuritis"

_viruses, 2023, doi:10.3390/v16010012_

Round 1

Reviewer 1 Report (Previous Reviewer 2)

Comments and Suggestions for Authors

Authors have addressed all my issues.

Author Response

Thanks for your comments and help in revising the manuscript. 

Reviewer 2 Report (Previous Reviewer 1)

Comments and Suggestions for Authors

The authors have provided summarization of the association between HSV-1 and VN. The manuscript is significantly improved. Some minor language issues as detailed below need to be addressed.

-Figure 3. 'IKK epsilon' instead of 'IKK'

Comments on the Quality of English Language

1. Line 110-115. The long sentence needs to be rewritten. Too much information is included.

2. Line 132, 'andlear' needs fixing.

3. Line 171, delete 'can'.

4. Line 199, 'play important roles'.

5. Line 217, 're' - 'reactivation'?

6. Line 231, what does 'hypothesis' mean?

Author Response

We have made revisions based on your comments. The following is our point-to-point  revisions to the comments.

Comment 1: -Figure 3. 'IKK epsilon' instead of 'IKK'.

Reply: Thanks for your comments. We change the 'IKK' with 'IKK epsilon' in Figure 3.

Comment 2: Line 110-115. The long sentence needs to be rewritten. Too much information is included.

Reply: Thanks for your comments. We have split this one sentence into two. The following is modified sentences.

-Line 110-114.

Arbusow et al. also reported the presence of HSV-1 DNA in human vestibular ganglia and nucleus vestibulitis, suggesting that the virus may migrate to the human vestibular labyrinth and cause inflammation. The inflammation in VN patients may lead to acute unilateral vestibular deafferentation and benign paroxysmal positional vertigo.

Comment 3:   Line 132, 'andlear' needs fixing.

Reply: Thanks for your comments. We replace 'andlear' with ' and vestibulocochlear'. The following is modified sentences.

-Line 131-132.

Additionally, connections between the facial and vestibulocochlear nerves occurred more frequently with the SVN than with the IVN.

Comments 4:  Line 171, delete 'can'.

Reply: Thanks for your comments. We delete 'can'. The following is modified sentences.

-Line 170-172.

VP16, as a capsid protein and a structural of the virus, plays a role in the structural components of the virus once HSV-1 enters the cell.

Comments 5:  Line 199, 'play important roles'.

Reply: Thanks for your comments. We add 'play'. The following is modified sentences.

-Line 197-200.

PUL36 and pUL37 guide the outer shell to move towards the position of the secondary envelope on the microtubule through interaction with the microtubule, and also play important roles in the transport of HSV from the nucleus to the periphery by binding to motor proteins.

Comments 6:  Line 217, 're' - 'reactivation'?

Reply: Thanks for your comments. We replaced 're' with 'reactivation'. The following is modified sentences.

-Line 215-217.

Several viral factors such as VP16 that may play a role have been identified, however, the theories and mechanisms underlying latent infection and reactivation are still in need of refinement.

Comments 7:  Line 231, what does 'hypothesis' mean?

Reply: Thanks for your comments. We replaced ' hypothesis ' with ' virus hypothesis '. The following is modified sentences.

-Line 229-232.

Rujescu et al. identified genome-wide associations with vestibular neuritis in four regions containing protein-coding genes that can be grouped into two functional categories: virus hypothesis and insulin metabolism, through a genome-wide association study.

This manuscript is a resubmission of an earlier submission. The following is a list of the peer review reports and author responses from that submission.

Round 1

Reviewer 1 Report

Comments and Suggestions for Authors

Zhao et al. wrote a review summarizing the viral life cycle of HSV-1 and HCMV, as well as their interactions with host physiological activities including innate/adaptive immunity. Overall the review is well-written and citations are appropriate. However, as the title focused on 'vestibular neuritis', the manuscript failed to provide sufficient summarization and discussion on the viral pathogenesis of the two herpesviruses related to this specific disease, which needs substantial revisions.

Other comments:

(1) The abstract looked more like an introduction than a summarization of the key points written in the main text.

(2) Figure 3 included both TLR and RLR signaling pathways but the authors only discussed viral-host interactions on TLRs. Furthermore, DNA sensing pathways such as cGAS signaling play essential roles in the antiviral activities against DNA virus, however, they were not mentioned in the manuscript.

Reviewer 2 Report

Comments and Suggestions for Authors

This is a review article entitled, “Cellular processes induced by HSV-1 and CMV infections in vestibular neuritis.” It is essentially a general review of the viruses, HSV-1 and HCMV. As such it is quite good in describing the membrane mediated events and certain aspects of host pathogen interactions such as, adaptive immune response, apoptosis. But it falls short in many aspects. Generally, it ignores the some crucial events in the nuclear replication cycles of both viruses. For example, figure 2 is overly simplistic. It occupies a whole page and conveys little. 

While the description of host response is better by comparison, the review does not go into the fact that both viruses are DNA viruses and as such are recognized by DNA sensors, such as cGAS, and induces innate immune gene expression through the STING pathway. Figure 3 totally omits this. 

No mention is made of ICP0. ICP0 is an HSV IE protein involved in overcoming several host restriction mechanisms including the degradation of PML. The review mentions the analogous activity for HCMV, IE-1, so this omission is not understandable.

No mention is made of VP16, which is also involved in the reactivaton of HSV from latency. Latency and reactivation are poorly covered, which is unfortunate, given that HSV establishes latency in nerve cells and this is supposedly a review on vestibular neuritis.

This review is an incomplete review of the viruses HSV-1 and HCMV. More importantly, it does not go into how any of what’s in the review may be related to vestibular neuritis. Vestibular neuritis is mentioned; 1 once in the Title, 4 times in the Abstract, 4 times in the Introduction, and 2 times in the 1 paragraph Conclusion. It is not mentioned in the body of the review. The review references 178 papers; only 1 has vestibular neuritis in its title. 

Reviewer 3 Report

Comments and Suggestions for Authors

In this article, Zhao and colleagues aimed to review the cellular and immunological effects of HSV-1 and HCMV in vestibular neuritis. I have to admit I was excited about this manuscript after reading the title, abstract and introduction since herpesviruses have been proposed as etiological agents for many different ear-related disorders, but the information remains spare and dispersed in the literature. I thought this would be the long overdue review of the existing data on the effects of herpesviruses in the middle and inner ear. However, after the introduction, the authors do not focus on hearing, or the vestibular and neurological system of the ear, but simply describe the general processes of viral entry, replication, and latency of HSV-1 and HCMV, and the cellular and immune responses to these viruses. There are already many other highly cited reviews of these notions in the literature, and I do not see the value that this manuscript adds over those existing reviews. The current manuscript is well written, although it would benefit from a better structure to present the ideas, which at times are presented in a very dense and disconnected way. For these reasons, I can not recommend publication. I encourage the authors to work on a more ear-centered or  neuron-centered review.